# A Community-Based Participatory Research Approach to Developing and Testing Social and Behavioural Interventions to Reduce the Spread of SARS-CoV-2: A Protocol for the ‘*COPAR for COVID*’ Programme of Research with Five Interconnected Studies in the Hong Kong Context

**DOI:** 10.3390/ijerph192013392

**Published:** 2022-10-17

**Authors:** Alex Molassiotis, Yao Jie Xie, Angela Y. M. Leung, Grace W. K. Ho, Yan Li, Polly Hang-Mei Leung, Hua Li Wang, Catherine Xiao Rui Chen, Danny W. K. Tong, Judy Yuen-man Siu, Joseph T. F. Lau

**Affiliations:** 1School of Nursing, The Hong Kong Polytechnic University, Hong Kong SAR, China; 2Health and Social Care Research Centre, University of Derby, Derby DE22 1GB, UK; 3Department of Health Technology & Informatics, The Hong Kong Polytechnic University, Hong Kong SAR, China; 4Department of Family Medicine and Primary Healthcare, Hong Kong West Cluster, Hospital Authority, Hong Kong SAR, China; 5Queen Elisabeth Hospital, Hong Kong SAR, China; 6Hospital Authority, Hong Kong SAR, China; 7Department of Applied Social Sciences, The Hong Kong Polytechnic University, Hong Kong SAR, China; 8Centre for Health Behaviours Research, Jockey Club School of Public Health & Primary Care, The Chinese University of Hong Kong, Hong Kong SAR, China

**Keywords:** COVID-19, coronavirus, community-based participatory research, social intervention, behavioural intervention, community intervention, health literacy, vaccine hesitancy, vaccination, early testing, prevention

## Abstract

Background: While a number of population preventive measures for COVID-19 exist that help to decrease the spread of the virus in the community, there are still many areas in preventative efforts that need improvement or refinement, particularly as new strains of the virus develop. Some of the key issues currently include incorrect and/or inconsistent use of face masks, low acceptance of early screening or vaccination for COVID-19, vaccine hesitance, and misinformation. This is particularly the case in some vulnerable populations, such as older people with chronic illnesses, ethnic minorities who may not speak the mainstream language well and children. The current protocol introduces a large programme of research through five interrelated studies that all focus on social and behavioural interventions to improve different aspects of community-related preventative indicators. Hence, the specific objectives of the overall programme are to (1) increase early testing for COVID-19 and promote the uptake of COVID-19 vaccines in the community (Study 1); (2) increase COVID-19-related health literacy and vaccine literacy and promote improved preventative measures in minority ethnic groups, chronically ill populations and caregivers (Study 2); (3) strengthen the public’s motivation to stay at home and avoid nonessential high-risk activities (Study 3); (4) decrease COVID-19 vaccine hesitancy (Study 4); and (5) enhance the adherence to COVID-19-related hygiene practices and the uptake of early testing in school children (Study 5). Methods: We will utilise a community-based participatory research (CBPR) approach in the proposed studies. All studies will incorporate an intervention development phase in conjunction with key community stakeholders, a feasibility study and an execution stage. A variety of self-reported and objective-based measures will be used to assess various outcomes, based on the focus of each study, in both the short- and long-term, including, for example, the 8-item self-reported eHealth Literacy Scale (eHEAL) and objective measures such as vaccine uptake. Discussion: Theory-driven interventions will address each study’s focus (e.g., social distancing, promotion of vaccine uptake, eHealth education, preventive measures and early detection). Improvements are expected to be seen in the outcomes of vulnerable and high-risk groups. Decreased infection rates are expected due to improved preventative behaviours and increased vaccine uptake. Long-term sustainability of the approach will be achieved through the CBPR model. The publication of this protocol can assist not only in sharing a large-scale and complex community-based design, but will also allow all to learn from this, so that we will have better insight in the future whether sharing of study designs can elicit timely research initiatives.

## 1. Introduction

Strategies to motivate, enhance and monitor personal behaviours for long-term COVID-19 prevention are an important challenge for healthcare workers, especially as the epidemic continues. Currently, the epidemic is characterised by invisible transmission chains, an increased number of asymptomatic carriers and community outbreaks. Many COVID-19 cases occur in the community, indicating poor compliance with social distancing regulations in certain Hong Kong (HK) populations. While the HK government has disseminated educational materials to the public, it is unclear how many people are fully aware of this information and why some do not accept the suggestions made. People with different cultural, professional or living backgrounds may have different reasons for non-adherence, and their opinions may change with changes in their external environment. Thus, a single fixed intervention may not be suitable to adapt to changing conditions. Quick, simple and flexible community-based health education projects would be the most appropriate choice to solve this issue. Given this emergent situation, multiple community-engaged efforts are needed to mitigate the spread of COVID-19. The key current issues in HK (and in many parts of the world alike) include the following.

The incorrect usage of masks (e.g., frequency and position) and low compliance with social distancing measures are notable in HK [1,2], indicating a need for improved education and guidance.

The HK population had a relatively low acceptance rate of COVID-19 early testing and vaccination measures. An attitudes and behaviours survey in 2021 showed that only 37% of the population would be willing to be vaccinated [3], although actual rates of vaccination are now high (84.3% with two doses as of 29 March 2022; https://www.covidvaccine.gov.hk/en/dashboard/ (accessed on 2 August 2022)) except in the older populations where rates are still below 40%. Merely educating people about the effectiveness and safety of the vaccine is not sufficient to alleviate concerns and it does not effectively address the causes of vaccine hesitancy. Rather, an open dialogue to disseminate information based on professionalism, transparency and up-to-date scientific evidence is necessary [4].

The World Health Organization asserts that the world is suffering from not only a pandemic, but also an ‘infodemic’ [5]. As a result of the pandemic, people are eager to learn about the ongoing situation from sources including the Internet and social media [6]. However, the influx of conflicting information from different sources can lead to uncertainty, distress and difficulties in making appropriate health-related decisions, particularly among certain cultural groups, such as South Asian communities [7].

Among the different social distancing measures currently being promoted, the public’s commitment to voluntarily stay at home and reducing the frequency of nonessential outings, particularly at times of outbreaks, can play a vital role in reducing community transmission. An analysis of mobility data from 130 countries showed that a weekly increase in compliance with stay-at-home measures by just 1% at the population level can prevent an average of 70 COVID-19 cases and seven deaths per week [8]. However, recent research indicates that staying at home during the COVID-19 pandemic is emotionally demanding for HK citizens [9], and thus, their motivation to voluntarily stay at home is likely to diminish as the pandemic persists.

## 2. Adoption of Self-Protecting Measures

As mentioned previously, only 37% of the HK population was willing to receive a COVID-19 vaccine in 2021 [3]. The percentage at the time was also low (40–63%) for healthcare professionals, such as nurses [10,11]. The reasons for vaccine refusal or hesitation include ‘suspicion on efficacy, effectiveness and safety’, ‘believing it unnecessary’ and ‘no time to take it’ [11]. To reduce the incidence of severe cases and mortality, it is important to achieve herd immunity using safe and effective vaccines. It is necessary to provide the public with ‘one-stop health education’ that covers self-protection from infection, early testing for COVID-19 and vaccination and post-vaccination procedures. As these behaviours are linked, failures at any link will increase the risk of infection.

Furthermore, social distancing is the most effective non-pharmaceutical intervention to prevent the spread of COVID-19 [12]. Despite the rapidly evolving pandemic landscape as vaccines become more widely available, social distancing will remain an essential containment strategy for the foreseeable future, due to the time required to ensure coverage and develop herd immunity; vaccine hesitancy and the ineligibility of key vulnerable groups (e.g., young children); and the emergence of vaccine-resistant virus variants. Additionally, the fact that the transmission pathways of severe acute respiratory syndrome coronavirus 2 (SARS-CoV-2) are now predominantly community-based underscores the need to strengthen the public’s engagement in practising effective risk-reduction behaviours. This includes their commitment to stay at home when possible and avoid nonessential high-risk social interactions (e.g., those conducted indoors, in crowded spaces, require mask removal (e.g., dining), and involve multiple households) that counter prevailing public health recommendations [13]. However, recent research indicates that staying at home during the COVID-19 pandemic is emotionally taxing for HK citizens [9]. Recent studies have shown that ‘safer-at-home’ public health advisories and the use of effective information campaigns are proactive, less costly, and have a greater positive impact on attitudes towards social distancing compared with government-imposed mandates [12].

## 3. Digital Health Literacy and COVID-19

COVID-19 is an example of a communicable disease in the community that requires citizens to respond quickly and work together. An electronic platform is the best option to disseminate health information to a massive population within a short time. Unfortunately, rampant misinformation is found on various Internet platforms, and WHO has asserted that the world is suffering not only from a pandemic but also from an infodemic [5]. During the pandemic, people are eager to receive health information from alternate sources, such as the Internet and social media. However, the influx of information from such sources can lead to uncertainty, distress and difficulty in making appropriate health decisions. Digital health literacy (DHL), the capacity to access, understand, evaluate and apply health information from electronic sources [14], is an important attribute that everyone should possess. Adequate DHL could enable people to process health information from the Internet and use appropriate means to deal with problems related to the pandemic [15]. Social media platforms (e.g., Facebook and WhatsApp) are the most common outlets for sharing COVID-19 information. Recent studies from our group in HK have shown that DHL is an issue facing people of all ages, especially ethnic minorities (EMs), people with chronic illnesses (PWCI) and professional and lay caregivers (CGs), during the COVID-19 pandemic [16,17]. During this pandemic, EMs and PWCI are particularly vulnerable groups. According to the 2016 Population By-census, 584,383 EMs lived in HK, representing 8% of the whole population. Five ethnic groups are dominant: Filipinos (184,081, 31.5%), Indonesians (153,299, 26.2%), Indians (6462, 6.2%), Nepalese (25,472, 4.4%) and Pakistanis (18,094, 3.1%). In 2013, 1,375,200 people (19.6%) in HK were living with at least one chronic condition [18]. PWCI have more health-related concerns, especially during a pandemic, and may have an increased risk of death from COVID-19. CGs (either staff in residential care homes or family CGs) take up the responsibility to obtain updated information about COVID-19 to support frail older adults. In view of the vulnerability of these three groups (i.e., EMs, PWCI and CGs) and the use of technology during the pandemic, it is crucial to cultivate DHL so that these groups can protect themselves and older adults from being infected. It is also crucial to support their access to and understanding and interpretation of COVID-19 information online and their decision-making regarding their behaviour (including hand hygiene, respiratory hygiene and receipt of vaccination).

## 4. Vaccine Hesitancy

Vaccine hesitancy is defined as ‘delay in acceptance or refusal of vaccines, despite the availability of vaccine service’ [19] and was named one of the top ten global health threats by the World Health Organization in 2019. Vaccine hesitancy is complex and context-specific, and it does not occur solely due to lack of information but is also determined by the various individual, group, and contextual factors [19]. Evidence indicates that the potential acceptance rate of COVID-19 vaccines among the public ranges from 50% to 80% globally, and a recent survey in HK found only a 40% rate of COVID-19 vaccine acceptance among healthcare workers [11]. Vaccines were made available to certain priority groups in HK in late February 2021. The COVID-19 vaccines are now available for almost all HK citizens (aged 3 to over 80) except for some populations who might have severe allergic reactions to vaccines. The latest evidence shows that over 85% of the HK population has received the second dose, and nearly 40% of the population has received the third dose of COVID-19 vaccines. Though more and more HK residents have completed their second or booster dose of a COVID-19 vaccine, worries about its safety and effectiveness still exist in certain groups of the HK population. In comparison to other age groups, older people (especially those over 70 years old) and children (under 11 years old) have a relatively low vaccination rate, with less than 40% of the older people (age over 70) receiving the third dose and only around 20% of children (age under 11) having received the second dose. Most of the deceased cases in HK are unvaccinated persons. Additionally, mutations in the virus causing COVID-19 might necessitate the development of new vaccine products in the future, leading to the need for regular vaccination. Additionally, vaccine hesitancy is a long-lasting phenomenon, and the vaccine hesitancy level fluctuates from time to time. People’s intention and motivation to receive the COVID-19 vaccines might decline when the community incidence of the positive diagnosed cases declines. Therefore, an effective intervention for reducing hesitancy and improving people’s motivation to be vaccinated against COVID-19 is needed [20], not only for the current urgent public health needs of achieving herd immunity but also for future long-term utilization in response to virus mutations and changes in people’s perception towards COVID-19 vaccines.

Various strategies at either the community or the individual level have been adopted to address vaccine hesitancy among the parents of children and other segments of the public. Specific programmes include communication or information tools for parents/healthcare workers, advocacy campaigns, and reminder-recall systems. Interventions usually focus on misinformation/safety issues, trust, religious/philosophical views, and perceived benefits [19]. Many previous interventions to reduce vaccine hesitancy targeted knowledge deficits. However, studies show that merely educating people about the effectiveness and safety of a vaccine is insufficient to address vaccine hesitancy effectively [4]. It is necessary to explore people’s personal views and reasons for vaccine hesitancy. A respectful dialogue communicating the most accurate information through professionalism, transparency, and up-to-date scientific evidence is necessary [4]. Motivational interviewing (MI) is an effective communication tool to address vaccine hesitancy among patients with various clinical conditions (e.g., human papillomavirus or prenatal care settings) [4,21]. MI sessions delivered at a maternity ward significantly increased the intention of mothers to vaccinate their infants [21]. MI adopts five core communication techniques (open questioning, affirming, reflective listening, summarising, and informing) to enhance patients’ internal motivation for attitudinal change by exploring and solving inherent ambivalence [4]. MI skills facilitate open discussion to address the specific concerns of an individual towards a vaccine. They can help engage the client without making them feel attacked, thereby motivating people’s own willingness to consider new or alternative views and solutions [4,21]. MI is widely used in substance abuse, eating disorders, and smoking cessation. MI skills have also been applied to help treat various conditions using advanced computing technologies, including app-based or Web-based platforms. With digital support and an e-health platform, services can reach a large group of users, save healthcare resources and time, and protect privacy [22].

## 5. Issues with Primary School Children

SARS-CoV-2, which causes COVID-19, is transmitted by contact with contaminated respiratory secretions or by inhalation of contaminated respiratory droplets. A large-scale outbreak in the USA showed that children are susceptible to COVID-19 infection and play an important role in its transmission, particularly when physical distancing and hygiene practices are not strictly observed [23]. While the current COVID-19 vaccination programme in HK covers children aged 3 and above, a study reported that children of 2–10 years and 10–16 years had higher prevalence of the UK variant (B.1.1.7) [24]. With the emergence of new variants of COVID-19, hygiene practices are essential for the prevention of COVID-19 transmission. The local government emphasises vigilance in hygiene, physical distancing and voluntary COVID-19 screening, with the latter facilitating early detection of COVID-19. As children generally exhibit less obvious symptoms of COVID-19 infection, or are asymptomatic, children’s participation in voluntary screening is essential COVID-19 to prevent the transmission of COVID-19. However, most promotional and educational materials on hygiene and voluntary SARS-CoV-2 screening are designed for adults, rather than children. Another problem is that self-collected deep-throat saliva, which is currently used for community COVID-19 testing, is difficult to obtain from children. Recent studies showed that pharynx gargle samples (throat wash) are good alternatives to self-collected nasopharyngeal swab specimens in terms of COVID-19–detection accuracy [25] and are easy to collect.

Educational virtual reality games (VRGs) are emerging technologies in teaching and learning. VR gives learners the illusion of being physically present in a real environment, which facilitates understanding of complex concepts and improves cognitive abilities such as memory and learning motivation [26]. Educational VRGs give instant feedback to users, which enhances learning and is important for behavioural change [27]. A literature review showed that the only existing VRGs related to hand hygiene are designed for healthcare workers [28]: there is no VRG-based educational intervention on hygiene practices designed for primary school schoolchildren.

## 6. The Community-Based Participatory Research (CBPR) Approach

Thus, we propose using a CBPR approach to design and implement health education projects to address the social dynamics of COVID-19 evolution. The HK society is highly developed, and there are a number of nongovernmental organisations (NGOs) and institutions that provide a diverse range of services and support to community members. Our projects will be the first to use the CBPR approach to integrate best community work practices with public health science to reduce the risk of COVID-19 in the HK community. CBPR allows the exploration of local knowledge and perceptions, and empowers people by considering them as agents who can investigate their own situations. Community input makes the project credible and enhances its usefulness by aligning it with what the community perceives as important social and health goals [7].

A simple, ‘one-size-fits-all’ intervention may not be sufficiently adaptive for the constantly evolving situation. Instead, simple flexible community-based health education projects may be more suitable to mitigate the spread of COVID-19. The proposed CBPR is a promising approach and has several advantages for reducing the risk of COVID-19 outbreaks in the community. In contrast to the positivist assumptions of traditional research approaches, CBPR is *‘a partnership approach to research that equitably involves, for example, community members, organisational representatives and researchers in all aspects of the research process and in which all partners contribute expertise, shared decision making and ownership’* [29,30]. CBPR focuses on the local relevance of public health problems and on ecological perspectives that address multiple determinants of health [31]. Partnering with community stakeholders can significantly facilitate translation into practice. CBPR entails: (1) targeted and suitable interventions, (2) flexible and adaptive solutions, and (3) sustainable approaches. During the process, the capacity of community stakeholders increases, and they become empowered to continue the work, even after the programme ends. Thus, CBPR-based health education projects motivate community participation, which increases their impact and reduces the effort and resources required from the government and research institutions. We propose a programme of research that includes five discrete studies/interventions targeting the community and vulnerable groups in HK. Based on the contexts of COVID-19 epidemic in HK, the five studies will establish partnership with different community stakeholders, to collaboratively design and implement a series of interventions for the target population, to address various research questions in the community setting, with an overarching aim to mitigate the COVID-19 risk in HK society (Figure 1).

The aims of this programme of research are to: (1) increase early testing for COVID-19 and promote the uptake of COVID-19 vaccines in the community (Study 1); (2) increase COVID-19-related health literacy and vaccine literacy and promote improved preventative measures in vulnerable groups such as minority ethnic groups, chronically ill populations and caregivers of older people (Study 2); (3) strengthen the public’s motivation to stay at home, comply with social distancing measures and avoid nonessential high-risk activities (Study 3); (4) decrease COVID-19 vaccine hesitancy (Study 4); and (5) enhance the adherence to COVID-19-related hygiene practices and the uptake of early testing in school children (Study 5). The publication of this protocol can assist not only in sharing a large-scale and complex community-based design, but also will allow all to learn from this, so that we will have better insight in the future whether sharing of study designs can elicit timely research initiatives. It can help also to offer a readily available design to implement immediately in other pandemic/endemic situations in the future.

The specific context of each study of this programme of research, named **Co**mmunity **Pa**rticipatory **R**esearch for COVID (COPAR for COVID), is explained below:

## 7. Study 1

AID (assistance, improvement, development): A community-based participatory research approach promoting the early detection of, and vaccination against, COVID-19 and enhancing the adoption of self-protective measures in HK

An impactful health education programme working with local community stakeholders together for improving the public’s attitude and behaviour towards COVID-19 prevention is needed. Therefore, in this study we plan to use the CBPR approach to establish an academic–community platform, collaborate with several community partners to design and implement a series of educational programmes for COVID-19 prevention in the community.

### 7.1. Aims

The ultimate goal of the proposed study is to mitigate COVID-19 risk in HK communities. Objective 1: To equip community partners with the knowledge and skills necessary to design and implement appropriate interventions for COVID-19 prevention. Objective 2: To carry out community-based education programmes for improving early testing for COVID-19, increasing vaccination rate and enhancing self-protection measures against COVID-19 in HK. Objective 3: To determine the most sustainable and suitable community education model and use it on a large scale across different communities. 

### 7.2. Plan of Investigation

A 4-year community-based health education project known as ‘AID (Assistance, Improvement, Development)’ will be implemented. The project will have four stages. Stage 1 and stage 2 are community mobilisation stages. Stage 1 will involve academic–community team building through administrative processes. Stages 2 is capacity building stage that mobilize the partners to enhance their ability for conducting community programmes. Stage 3 will involve the design and implementation of the interventions using a cluster randomised controlled trial (RCT), and Stage 4 will be the model-enhancing stage, in which the achievements and experiences from the previous three stages will be summarised and disseminated to the wider public.

#### 7.2.1. Subjects

In accordance with the CBPR approach, two kinds of people will be involved in our project: research partners and study participants. Research partners will help to recruit study participants. The study participants will be land-based non-institutional HK residents. Three types of participants will be recruited: adults from the general population, high-exposure groups, and middle school students and their parents. General adults are the majority of the HK population, their responses and feedbacks to the health education programme are representative and important. The high-exposure groups will consist of those more frequently exposed to the virus, such as truck, taxi and bus drivers; fitness instructors; restaurant waiters; and employees of cold-chain logistics companies. These individuals may have a higher risk of being in contact with virus carriers, becoming infected and transmitting the virus to other people. School students are another target group requiring particular attention for infection protection, and their parents will also be involved in the project. With these considerations, we will choose research partners in three main sectors: (1) local community NGOs, (2) company and business employers and (3) teachers in middle and high schools. Sector in this study means the organization that we target to collaborate with. The research partners will be invited from these sectors. Research partners from these sectors will help in the recruitment of participants and expediently carry out the intervention programmes. For participants recruited in sectors (1) and (2), the inclusion criteria will be: (1) HK residents aged 18 years or above and (2) agreement to participate in the study and provide written informed consent. Participants in sector (3) will consist of students aged 12 to 18 years, studying in public secondary schools in HK, and their parents. Those who cannot provide written informed consent (or assent for the younger students) will be excluded from the study. 

Sample size: As a cluster RCT, the randomization is conducted in the cluster level. In our study, each cluster is a group of participants recruited by the same research partner. In our study, based on our simulation modelling [32], a 20% increase in the vaccination rate could have significant social benefit. With conventional assumptions of a two-sided 5% significance level and assuming an intra-cluster correlation of 0.01, we would have 86% power to detect a difference of 20% if there are 50 participants in each cluster and 8 clusters in each sector [33]. Thereby 3 sectors will have 3 × 8 = 24 clusters and 1200 participants in total. For each sector, 8 clusters (400 participants) will be evenly divided into either the intervention group or the control group. When we used the same method to calculate the sample size for detecting 20% difference in the percentage of early testing (another primary outcome), the sample size was smaller than the previous one. We thereby consider the former one as our final sample size, as it could ensure the power of the study.

#### 7.2.2. Methods

Following the characteristics of the CBPR approach, this proposed project will have the following four stages:

##### Stage 1: Establishing the Collaboration Platform with Local Community Stakeholders

We will officially invite target NGOs, middle schools, and companies/businesses to join our project as research partners. They will be involved in forming the collaboration platform for the project. Collaboration will occur at various levels. Working groups consisting of centre-in-charge/frontline workers will be designated by their managers. They will work on the design, implementation and evaluation of the intervention through the administration of questionnaires, participation in process evaluations, the production and dissemination of professional toolkits and the promotion of educational programmes to the public. Once the collaboration platform is formed, steering committee meetings will be held every 3 months to monitor the progress of the intervention programmes, discuss the challenges and barriers encountered in the implementation of the programmes, and identify solutions for any problems encountered.

##### Stage 2: Pilot Study and Capacity Building for Research Partners: Train-the-Trainer (TTT) Workshops

Design and setting: stage 2 includes two parts, a pilot study for participants and a mixed-methods study for the research partners. The purpose of the pilot study is to determine the most optimal components of the ‘core intervention package’ which will be delivered in the formal cluster RCT. We will conduct a single arm pre-post experimental study among 50 participants in the community, to assess the effect of intervention components and assessing whether the intervention component effects vary with the individual’s current context. The pilot study will last for 4 months. The evidence obtained from the pilot study will be introduced to the workshop participants, to facilitate the design and implementation of their own programmes later. The research partners will designate their centre-in-charge/ frontline staff to attend the TTT workshops. Each collaborator will designate 2–3 staff members. We expect that at least 48 individuals will attend the training workshops. Two rounds of workshops will be conducted within 5 months. The staff will be divided equally into the two rounds of training. 

Training content: The workshops will have 3 sessions, and each session lasts for 2 h. The content will include: (1) basic cluster RCT design and CBPR knowledge; (2) key knowledge of COVID-19 prevention and protection, and the benefits of early testing and vaccination; (3) the use of the PROCEDE-PROCEED model [34] to assess the health needs and develop the community-based education programmes; (4) the use of a knowledge-attitude-practice (KAP) model [35] to evaluate the effectiveness of the health education intervention; (5) the content of the core intervention package; and (6) the suggested channels and approaches for intervention implementation. 

Evaluation: Pre- and post-programme evaluations will be implemented. Before attending the workshop (T_0_), all participating staff will be required to complete a questionnaire asking about information related to the COVID-19 epidemic and the content of the training workshop, to evaluate their knowledge, attitude and intention to implement the community education programmes. Similar assessments will be performed soon after completion of the training workshop (T_1_) and after implementing the intervention programmes (T_2_). Furthermore, two focus group interviews will be conducted for twenty randomly selected trainees, to understand their views on the training workshop in terms of the content, method, materials, their level of satisfaction, experiences and views on implementing the education programmes, and suggestions for improvement.

##### Stage 3: Implementing the Community-Based Health Education Programmes

Design and setting: stage 3 is a multicentre cluster RCT. The intervention programmes will be implemented by the research partners. 

Randomisation and concealment: The participants recruited by one research partner will be considered as one cluster. A research assistant will randomly assign the clusters from the NGOs, Schools, and Companies to either the intervention group or the control group, respectively. Concealment of the allocation sequence to the investigators will be done until assignment occurs. After allocation, there will be four clusters in both the intervention and control groups from the NGOs, schools and companies, respectively. This will result in 3 × 4 = 12 clusters for each group (intervention/control). An active wait-list control group will be used instead of a standard control group to help motivate the collaborators to implement the programme and to avoid a high participant dropout rate. Intervention: The core intervention package will potentially include the following four components. (1) A manual of self-protection against COVID-19 infection. The main content will include hand washing, mask wearing and social distancing guidelines. (2) Early testing. Participants will be trained on how to recognize the early symptoms of COVID-19 and appropriate practice to take in a response, and they will be provided with testing resources. The COVID-19 Rapid Antigen Test (RAT) will be provided to participants as an incentive to facilitate self-testing at home. This RAT is a lateral-flow immunoassay designed for the qualitative detection of SARS-CoV-2 antigens in nasal swab samples. (3) Knowledge of vaccines and their benefits and resources for vaccination. (4) Post-vaccination follow-up. Participants will be trained on how to recognise vaccine side effects and to provide further protection measures, and they will be provided with resources for consultation and treatment. The research partners will design the community-based education programmes (based on the core intervention) after the TTT workshop. They will be required to submit a brief standardized proposal to the academic investigators for review and approval, to ensure that all intervention programmes have the same core intervention contents and can be implemented appropriately. The development and implementation of the intervention will follow the procedures outlined in the PROCEDE-PROCEED model (Figure 2). Research partners will be able to use any reasonable strategies, such as social media platforms, information technology, posters, leaflets, and videos, to implement the programmes. Each programme will last for 3 months. The active wait-list control group will undergo parallel assessment at the same time points as the intervention group, while the core intervention contents will be delivered after 3 months. We expect that the 24 intervention programmes will be completed within 2 years. Evaluation: Evaluations will be conducted at baseline (T_0_), middle term (T_1_, 1.5 months), and at the end of the trial (T_2_, 3 months). Participants’ knowledge, attitudes and behaviours towards COVID-19 prevention will be evaluated using a self-designed questionnaire, based on the KAP model. In addition, a process evaluation that determines whether the programme is being implemented as intended will be performed as follows. Every month, the academic investigators and the collaborators will review the progress of each programme together. Information on the attendance rate of each programme and feedback from program participants and research partners will also be evaluated qualitatively. A flow chart of the all the data collection steps is shown in Figure 3.

##### Stage 4: Health education model enhancement and promotion

An agent-based simulation model that considers human mobility in HK will be developed [36]. The model will be trained using data from the 24 community-based intervention programmes, to establish the most optimal community-based health education model(s) for COVID-19 prevention. We assume different intervention programmes will cause different changes in the vaccine uptake patterns among participants. Each intervention programme’s result (e.g., the vaccine uptake number) will be collected for comparison purposes. Then, the programme with maximal vaccine uptake and minimal healthcare requirements (e.g., number of infections detected) will be chosen as the most optimal strategy. Once the simulation modelling is completed, a toolkit book that contains (1) training workshop content and (2) details on the implementation and assessment of the most optimal health education model(s) will be produced and disseminated to additional NGOs, companies and schools. A health education pamphlet containing relevant COVID-19 information and the successful experiences of our health education programmes will be disseminated to the public via the research partners within one year. We expect that 200 NGOs, companies and schools will be approached, and 20,000 HK residents will receive the pamphlets. In addition, several press conferences will be held to share our findings and experiences with the public.

#### 7.2.3. Data Type, Data Processing and Analysis

Stage 1 will involve administrative processes for building the collaboration platform. No data will be collected at this stage. During Stage 2, quantitative data will be collected using a self-administered questionnaire. Qualitative data will also be collected through focus group interviews. The transcription of data will be done after the focus group interviews. During Stage 3, all data generated will be quantitative. The primary outcomes will be vaccination rates. It will be calculated as the number of participants who have been vaccinated over the total number of participants. Each participant’s vaccine uptake will be recorded by asking the participants to provide their COVID-19 vaccination record in the Vaccine Pass, or the paper vaccine certificate delivered by the HK government. The secondary outcomes will be the amount and frequency of early testing, the levels of acceptability for vaccination and early testing, and the changes in knowledge, attitude and behaviour towards COVID-19 protection and prevention, as determined by a self-administered questionnaire. The questionnaires will be designed based on the KAP model. The validity and reliability of the questionnaires will be assessed in advance.

During Stage 2, quantitative data from the pre- and post-programme evaluations will be analysed using a paired Student’s *t*-test for continuous variables and McNemar’s test for categorical variables. Qualitative data from the focus group interviews will be first organised by an experienced qualitative researcher. The transcripts will be coded and analysed in a systematic and iterative manner. Finally, themes and subthemes will be generated. During Stage 3, cluster RCT data will be analysed according to an intention-to-treat (ITT) principle. All participants who attend baseline assessment will be included in the ITT analysis. Within groups, changes at T_1_ and T_2_ time points will be compared with baseline (T_0_) data using a paired Student’s *t*-test or McNemar’s test, as appropriate. To compare the between-group differences in mean change from baseline to 3 months, as well as testing the time × group interaction effects between groups, linear mixed models will be adopted for analyses. The variables that are significantly different between groups at baseline will be used as covariates in the linear mixed models. Binary logistic regression models will be also used to examine the association between interventions and the main outcomes. Odds ratios will be calculated to determine the magnitude of the intervention effect.

## 8. Study 2

Digital Health Literacy on COVID-19 for All: Co-creation and evaluation of interventions for ethnic minorities and Chinese people with chronic illnesses in Hong Kong

This is the first DHL intervention to be developed with a community-based participatory research approach [37] and to be co-created with the stakeholders. It is essential to build up everyone’s capacity to access, understand and interpret the health information they receive and to make appropriate health decisions (i.e., DHL). Once they learn the skills or improve their ability to search for reliable health information and evaluate it in a critical manner, they can apply such skills and abilities to other health issues.

### 8.1. Aims

To develop and assess the efficacy of DHL interventions on self-protection of COVID-19 and promoting COVID-19 vaccination among EMs, Chinese PWCI and CGs in HK. *Objective 1.* To co-create DHL interventions with these three groups susceptible to COVID-19 and meet their specific needs in DHL to prevent infection. *Objective 2.* To assess the efficacy of the DHL interventions on eHealth literacy, vaccine literacy and actions taken for COVID-19 prevention.

### 8.2. Plan of Investigation

This 4-year research program will consist of three interrelated but independent studies of three specific populations with similar study designs. Study A will target five EMs in HK (Filipinos, Indonesians, Indians, Nepalese and Pakistanis), whilst Study B will target Chinese PWCI and Study C will target CGs (including health workers in residential care homes and family CGs of frail older people). Each study will have three phases. Phase 1 will involve the use of a community-based participatory research approach to involve stakeholders in the development of the interventions; Phase 2 will involve testing the feasibility and acceptability of the interventions; and Phase 3 will involve examining the efficacy of the interventions using a randomised controlled trial design.

#### 8.2.1. Subjects

Phase 1. Thirty EM adults, 30 PWCI and 30 CGs will be recruited for focus group interviews before the development of the interventions begins. Another 10 EM adults (2 from each ethnic group), 10 PWCI and 10 CGs will be invited to participate in cognitive interviews. Phase 2. Twenty EM adults, 20 PWCI, 20 CGs and 6 moderators will be invited to participate. Phase 3. One hundred fifty-six EM adults, 156 PWCI and 156 CGs will be randomised to the intervention or the control group.

We estimated the sample size for the trial of each proposed study based on a previous eHealth Literacy study that reported a small-to-moderate effect size (d = 0.245) [38]. For the proposed study, with five repeated measurements over a 6-month period, a total sample size of 108 subjects (i.e., 54 per group) is required to achieve a power of 90% when using a repeated-measures analysis of variance to detect a significant between-group interaction effect at 5% significance (G*Power 3.1). Furthermore, assuming an attrition rate of 30% [39], 156 subjects (78 per group) will be recruited for each study.

#### 8.2.2. Methods

A community-based participatory research approach [37] will be adopted in this study by inviting stakeholders (EM adults, PWCI and CGs) to participate in various phases of the study. This participatory method can provide opportunities for relevant stakeholders to share their perspectives on and knowledge of a particular topic (in this study, their ability to access, understand and interpret health information from the Internet or social media) with the research team and contribute their experiences to the development of learning objectives and materials in the DHL interventions. In this collaborative research process, the stakeholders will work with researchers to identify issues that are critical to their communities; identify resources and solutions; and develop, implement and evaluate interventions. We intend to engage the stakeholders in various research processes to understand and address their concerns. The materials of the interventions will be developed ‘*in context*’, i.e., where stakeholders easily find them relevant to their daily lives. This co-creation of new knowledge by stakeholders and researchers will increase the uptake of the research outcomes and adoption of the DHL interventions.

#### 8.2.3. Study Design

Phase 1. Each qualitative study will include three to five focus group interviews and five cognitive interviews. In the focus group interviews, the stakeholders will be encouraged to indicate their problems and concerns about DHL, propose resources/solutions and formulate a plan. Version 1 of the interventions will be based on these comments. Cognitive interviews will be conducted with the stakeholders when the version 1 interventions are available. Here, we will ask the stakeholders to go through the interventions step-by-step to evaluate their flow and content.

Phase 2. A feasibility and acceptability study with individual interviews will be conducted to collect views from 20 stakeholders and two moderators. The moderators will be the trained workers who facilitate online group discussions on social media platforms.

Phase 3. This will be a two-arm, prospective, parallel-group (1:1), randomised controlled trial.

Settings. Participants will be recruited from non-governmental organisations that provide services to EMs, PWCI and CGs.

Interventions. ‘Self-determination Theory’ will be used to guide the development of the intervention. That is, we address ‘autonomy, competence and connection (or relatedness)’ when we develop the intervention. While we are inviting the stakeholders (EM, PWCI, CG) to co-create the intervention with us, we ascertain they feel in control of their own behaviours and goals (whether they are determined to do the best to prevent COVID-19). We will also support them to gain mastery of tasks and learn different skills. Additionally, we will make sure the stakeholders experience a sense of belonging (attachment) to the materials we develop. The interventions will be developed based on the Best Practices for Digital Health Literacy [40], WHO health literacy toolkits [41] and the findings of the interviews in Phase 1. The content to be covered in the intervention will include news about vaccines; alerts about COVID-19 preventive measures; guidelines on social distancing, COVID-19 screening test arrangements, vaccination and compulsory quarantines; COVID-19–related law enforcement; and self-monitoring of COVID-19 symptoms. The interventions will be designed for three social media and communication platforms (a closed Facebook Social Learning Group, YouTube and Viber/ WhatsApp). Facebook and YouTube were chosen because they are the top two social media platforms in HK (used by 82% and 81% of users in HK, respectively) [42]. The ‘Social Learning Group’ in Facebook is a closed system that protects participants’ privacy, and it serves as the key platform for learning the short-clip videos our group developed for a different study [43], including step-by-step instructions from the moderators, eight online exercises and eight interactive sessions. The videos will be co-developed with representatives of the respective ethnic groups in six languages (English, Indonesian, Hindi, Nepali, Urdu and Cantonese), uploaded to YouTube and shared in the Facebook Social Learning Group. Development of the intervention will also consider the latest situation of COVID-19 regarding the content and the mode of delivery of preventive measures and vaccination recommendations. PWCI and CGs require different types of health information; for example, PWCI need information about the contraindications of vaccination to the long-term drugs they are taking, whilst CGs need information about the latest policies regarding visit restrictions or vaccination arrangements for residential care homes. Both groups will be invited to participate in the development of the intervention.

Procedures. This online learning programme will be operational for 6 months (eight weekly sessions in months 1 and 2 and three interactive booster sessions in months 3 to 6). Before using Facebook, a 1 h virtual or face-to-face training session will be provided to the participants to familiarise themselves with all of the features on Facebook and the privacy settings. Communication tools that enable end-to-end message encryption, such as Viber or WhatsApp, will be used for communication within small groups (six to ten participants supported by a trained moderator). Viber is the most common communication tool used by the Nepalese population, and WhatsApp is often used by Pakistani, Filipino, Indonesian and Chinese populations. Virtual bi-weekly group meetings (30 min each) will be held via group video calls in WhatsApp or Viber. In a pilot study [31], the participants recommended that the learning period be extended to 8 weeks plus some booster sessions in subsequent months. The fidelity of the intervention will be assessed by a trained research assistant who retrieves the record in the backend of the Facebook Social Learning Group and assesses the use of the materials by the participants, the duration of interactions among the participants and with the moderator, and the frequency of the use of materials in discussion. The process of the proposed study is shown in Figure 4.

#### 8.2.4. Data Types and Data Analysis

Phase 1. Qualitative data will be collected from focus group interviews and cognitive interviews, which will be transcribed verbatim. Phase 2. Two more sets of qualitative data will be collected from the stakeholders and moderators via interviews. Phase 3. Five measurements will be made: baseline, after the 8-week intervention (Month 2), after the 1st booster session (Month 3), after the 2nd booster session (Month 4) and after the last (3rd) booster session (Month 6). Primary outcome: eHealth Literacy Efficacy, measured by the eHealth Literacy scale (eHEAL), an 8-item self-reported efficacy scale scored on a 5-point Likert scale [14]. Secondary outcomes: Digital Health Literacy Instrument (DHLI) [44] and vaccine literacy (which includes intention to receive the COVID-19 vaccine) [38]. Knowledge and health behaviours related to COVID-19 prevention and the participants’ engagement in all activities will be assessed via Facebook Insights. Knowledge will be measured by 12 true-or-false items with a total score ranging from 0 to 12; attitudes towards COVID-19 will be measured by 2 yes-or-no items with a total score ranging from 0–2, and self-protection behaviours will be measured by 3 yes-or-no items with a total score ranging from 0 to 3. The self-protection behaviours are self-reported in three items: (1) going to crowded places, (2) whether wearing face masks when outside their home during the week before taking the survey, (3) whether washing their hands for 20 s each time when they return home or are in contact with another person) in yes-or-no answer [45,46]. Participants can report their behaviours and other outcomes in a link (that leads to an online survey in Qualtrics) in the Facebook.

Data analysis. Phases 1 and 2. The qualitative data analysis will occur in two steps. In the first step, a researcher will read through the transcripts and develop a codebook with definitions. Another researcher will then review the transcripts and add emergent codes as appropriate. The codebook and the definitions will be refined via discussion between the academic researchers and stakeholders. The second step of the analysis will focus mainly on the identification of themes and subthemes and on obtaining consensus across the team. Phase 3. To examine between-group differences in changes in the outcomes, we will perform separate linear mixed-effects models for each of the continuous outcomes (eHEAL, DHLI, vaccine literacy, knowledge, attitudes, and practices towards COVID-19) and three independent variables: ‘Time’, ‘Group’ and their interaction term ‘Time × Group’. A significant result of the interaction term (Time × Group) will indicate a differential change in the outcome variable between the two groups.

## 9. Study 3

#StayOkayHK: A universal public health initiative to promote stay-at-home during COVID-19

This project will develop and test a novel universal public health initiative (#StayOkayHK), incorporating a public health communication campaign, health education and promotion, and participatory social media strategies to encourage the HK public to stay at home to reduce the risk of SARS-CoV-2 transmission.

Participatory social media approach as a novel strategy to engage the HK public. It is estimated that 83% of HK citizens participate in social media and they spend an average of 13 h per week on these platforms [47]. Therefore, social media-based public health campaigns can be an effective strategy to engage citizens across sociodemographic groups. Further, the medium-long-term management of the pandemic and the changes it imposes on social norms and life rules call for additional strategies that empower the community to practice safe behaviours collectively and proactively. Therefore, we argue that adopting a participatory approach to engage the public in the content creation process and exchanging user-generated content will confer additional impact on the dissemination, uptake, and maintenance of COVID-19 public health campaigns. Indeed, research shows that user-generated content (e.g., videos and comments) enhances social identification, self-efficacy in performing health-promoting behaviours, and source credibility of information in health campaigns [48]. Moreover, it cultivates a positive affect and provides opportunities for creative self-expression in health-promotion initiatives [49]. Although no such public health campaign has been implemented in HK, evidence suggests that HK citizens have high levels of trust and willingness to peruse and share experienced-based health information on social media platforms [50,51].

Theoretical framework. This proposed public health initiative (#StayOkayHK) will be developed in accordance with Protection Motivation Theory (PMT) [52], which posits that environmental or intrapersonal sources of information about a health threat, such as communication campaigns, verbal persuasions, and the media, initiate the following two cognitive processes: (1) threat appraisal, comprising perceived vulnerability to and severity of the health threat and (2) coping appraisal, comprising perceived self-efficacy and response efficacy of health-protective behaviour. The threat and coping appraisals combine to exert a positive influence on an individual’s protection motivation (i.e., their motivation and intention to engage in health-protective behaviours), and these relationships are further counterbalanced by the costs to engage in health-protective behaviours and the rewards from engaging in health-compromising behaviours. PMT was selected given its (a) specificity to the context of the present investigation on health communication and information exchange to motivate health-protective behaviours for a specific health threat; (b) parsimony in guiding the design and practical implementation of a small set of core intervention components with high predictive value, and (c) strong evidence base when applied in the context of compliance with COVID-19 risk-reduction strategies [53,54]. Thus, the present project does not aim to test the principles of PMT, but rather, to directly apply its core tenets to this proposed initiative.

### 9.1. Aims

We aim to strengthen the public’s intention and motivation to stay at home and avoid nonessential high-risk social interactions to reduce the risk of community-based SARS-CoV-2 transmission, particularly during outbreaks and new waves of confirmed cases. The primary objectives are to strategically target increasing the public’s (a) threat appraisal of COVID-19 and (b) coping appraisal of staying at home. This is expected to enhance the public’s motivation and intention to stay at home. A secondary objective is to promote psychological well-being through participatory social media activities.

### 9.2. Plan of Investigation

#### 9.2.1. Subjects

This universal public health initiative will target all HK citizens. The campaign materials will be accessible to all via the internet, and the campaign will be further promoted in outdoor and digital advertisements. Citizens aged 18 years and above may voluntarily register for the campaign to receive content updates and complete baseline and ongoing surveys.

#### 9.2.2. Methods

This 4-year project will include 6 months of preparation and content/platform creation, 36 months of intervention and 6 months of summative evaluation and report writing. A public health initiative titled “Stay Okay HK” will include the following three core components: (1) a universal health communication campaign for #StayOkayHK, (2) a consolidated Web-based platform providing ideas for home-based activities and educational resources on COVID-19 risk reduction and health promotion and (3) participatory social media strategies to engage and motivate the public to stay at home.

The #StayOkayHK communication campaign will be developed using evidence-based COVID-19 communication strategies and the principles recommended by the World Health Organization for effective communication [55]. Health communication materials will be designed in accordance with the core tenets of PMT, and will be developed with creative and marketing agencies using focus groups representative of the general public. A campaign website will house all campaign information and activities linked to social media platforms (i.e., Facebook, Instagram, YouTube). Links to the campaign website (i.e., QR code and URL) will be clearly marked on all health communication materials and advertisements. Additional health education and promotion content will be provided by leveraging resources from existing government-led health initiatives (e.g., change4health, Joyful@HK, Shall We Talk, YouthCan). Original video content on home-based activities will be developed to further enhance the public’s stay-at-home self-efficacy, and reduce perceived cost of staying at home (e.g., boredom) and rewards of participating in nonessential high-risk social activities. The online campaign platforms will be opened for Beta testing two weeks before campaign rollout. The intervention will be delivered in two consecutive phases as described below.

Phase 1: Active Campaign (12 months): In Phase 1, #StayOkayHK health communication materials will be disseminated in a territory-wide campaign via different media and advertising channels (e.g., print, digital, and social media). Targeted promotion for vulnerable groups (families with young children or elders, youths, residents of small or subdivided housing) will also be carried out via social and advertising media (e.g., paper and outdoor advertisements; distribute promotional materials at markets, transit hubs, estate offices), and through community service agencies. Original videos and social media content on stay-at-home activities will be uploaded weekly onto campaign platforms. This aligns with a life-cycle approach, which is necessary to develop and sustain an online community for health promotion [56]. The initial batch of social media posts and videos will focus on four areas. The first two foci, (1) meal preparation and (2) physical fitness, will strategically provide stay-at-home alternatives to common nonessential high-risk social interactions (i.e., dining at restaurants, working out at a gym), thereby reducing the perceived rewards for engaging in those health-compromising behaviours. Foci (3) home improvement and (4) mental wellness will aim to maximise the enjoyment of time spent at home, thereby reducing the perceived cost of staying at home. Other focal areas (e.g., preserving harmony, caring for children and elders) may be created based on user feedback. To do so, citizens registered for the campaign will be able to ‘like’ and provide feedback on the website content. They will also be encouraged to create their own social media content on home-based activities inspired by the campaign (e.g., at-home exercising) and post them on social media with the campaign hashtag ‘#StayOkayHK’.

Phase 2: Social Participation and Campaign Maintenance (24 months): In Phase 2, three sets of activities will be carried out. First, selected content created by the public and discoverable on social media platforms with the campaign hashtag will be reposted on the campaign website at the beginning of each month. Reposted content in each focal area that receives the most views/likes/shares on the campaign website at the end of each month will receive a small incentive. Second, a stay-at-home-themed social media challenge tailored to the HK context will roll out every 3 months (eight challenges in total). A random lucky draw will be conducted 3 months after each rollout and 30 participants will receive a small incentive. Third, campaign maintenance activities (e.g., targeted promotion, advertising and new videos) will take place every 6 months to sustain momentum and public participation. To keep the health communication messages up-to-date and relevant, additional health education and promotion content or relevant public health messages will also be disseminated as part of the campaign maintenance activities. Together, these activities are expected to amplify campaign reach, maintain public interest and awareness of the #StayOkayHK initiative and harness the power of the community to promote citizens’ motivation and intention to stay at home.

Study design. A repeated cross-sectional non-comparative design will be used to document campaign uptake, process and impact outcomes. A single-group pretest–posttest design will be used to assess intervention effectiveness.

Data and analysis. Ongoing analytics and campaign-focused metrics will be collected in real time on the campaign website (e.g., hits) and social media platforms (e.g., views/likes/shares). These data will be compiled and reviewed at 3-month intervals (process evaluation) and at the completion of the project (impact evaluation). Descriptive statistics will be used to present process and impact metrics on reach, adoption, implementation and maintenance. At intervention completion, a representative single-frame (mobile phone) random digit-dial telephone survey of HK adults aged 18 years and above will also be conducted to provide further evidence of campaign reach and adoption. Respondents will provide a dichotomous response (i.e., yes/no) to whether they have heard of the #StayOkHK campaign, accessed the campaign platform, perused any campaign content, and participated in any social media activities; they will also provide information on their degree of usage and satisfaction if they gave an affirmative response to the above questions. Based on the estimated adult population of Hong Kong in mid-year of 2020, a conservative estimate of 50% prevalence of exposure to the campaign, precision of 3%, and 95% confidence interval, the minimum sample size required for this telephone survey is 1068 participants.

To assess intervention effectiveness, campaign registrants will provide basic demographic information (i.e., age, sex, household characteristics, employment status, social media account, contact information) during registration (T0) and complete self-reported measures of their (a) stay-at-home motivation, intention and past-week behaviour; (b) threat and coping appraisals, measured using adapted instruments deployed in recent studies that tested PMT in the COVID-19 context [53,54]; and (c) depressive, anxiety and stress symptoms, as measured using the 21-item Depression Anxiety Stress Scale [57]. Follow-up data will be collected at the end of the active campaign (T1), and every 6 months thereafter until the end of the intervention (T2, T3, T4, T5). Invitations to complete the surveys will be delivered via the participants’ social media accounts, phone calls or email, based on the contact information provided at registration. Up to five attempts will be made before the participant is deemed lost to follow-up for that time point. To assess effectiveness, logic checking and missing data analysis will be performed. Missing data will be handled based on the degree and mechanism of missingness (i.e., deletion or imputation). Generalised estimating equations will be used to model within-subject longitudinal data to estimate population-average changes in stay-at-home motivation, intention and past-week behaviour; COVID-19 threat appraisal; stay-at-home coping appraisal; and psychological well-being (depression, anxiety, stress) across respondents over time. A minimum sample of 515 campaign registrants will be targeted to achieve 80% power and alpha of 0.0042 (to keep the overall type 1 error at 0.05 for 12 outcomes) with small intervention effect (Cohen’s d = 0.2) across timepoints for 6 repeat measures with an assumption of a first order autoregressive covariance pattern and a base correlation of 0.30.

## 10. Study 4

AI-driven Vaccine Communicator: The impact of a Web-based psychoeducation programme with a motivational AI-based digital assistant on COVID-19 vaccine hesitancy in Hong Kong’s population

To address the widespread hesitancy among HK residents toward COVID-19 vaccines, we will develop a Web-based psychoeducation programme, ‘AI-driven Vaccine Communicator,’ driven by artificial intelligence (AI) techniques based on the Vaccine Hesitancy Determinants Matrix Model and MI communication skills and concepts (theoretical underpinnings) [19].

### 10.1. Aims

Our proposed study will aim to develop and evaluate the effectiveness of a Web-based psychoeducation programme to address COVID-19 vaccine hesitancy, ‘AI-driven Vaccine Communicator’ (including educational materials, animations of vaccine research and development, and an MI communication skills-based AI, digital assistant). Specifically, we will address three objectives in sequential stages.

Objective 1. To gain an in-depth understanding of the public’s perceptions of/concerns regarding COVID-19 vaccines across different ethnic and socio-economic groups, and develop the Web-based psychoeducation programme ‘AI-driven Vaccine Communicator’.

Objective 2. To conduct a randomised controlled trial to test the effectiveness of the ‘AI-driven Vaccine Communicator’ among members of HK’s population who are hesitant about COVID-19 vaccines.

Objective 3. To perform a pragmatic service evaluation of the ‘AI-driven Vaccine Communicator’ and its long-term significance in response to future mutations of the virus causing COVID-19 and the expansion of the vaccination priority groups.

Our goal is to standardise our intervention so that it can serve as an effective toolkit for clinicians/healthcare providers to increase HK residents’ motivation to vaccinate and to ensure that the programme can be adapted to viral mutations and newly developed vaccines in the medium/long term. The project will involve the active participation of the community, including the development, evaluation, and promotion of our intervention programme as guided by the CBPR approach.

### 10.2. Plan of Investigation

#### 10.2.1. Subjects

Our target population will be Hong Kong residents aged ≥18 who show hesitancy toward COVID-19 vaccines. Participants will be defined as COVID-19 vaccine-hesitant if they have no plan for a date to receive the available COVID-19 vaccines in either 1st, 2nd, or 3rd dose [19], or completed three doses of COVID-19 vaccines not based on their willingness and are hesitant for a future 4th dose or regular vaccines).

#### 10.2.2. Study Methods

Our study will be conducted in three stages. In Stage 1, we will perform an exploratory sequential mixed-methods study with an online survey followed by an exploratory qualitative study. In Stage 2, we will conduct an RCT to test the efficacy of the ‘AI-driven Vaccine Communicator’ relative to being self-informed about COVID-19 vaccines. In Stage 3, after the RCT (Stage 2), we will adopt and implement the intervention as a service evaluation to a wider HK population.

#### 10.2.3. Study design

Stage 1: HK residents of any ethnic and socio-economic background, who are at least 18 years of age and without difficulties in understanding the online survey, will be eligible to be participants. It is estimated that more than 20% of the population will show hesitancy toward COVID-19 vaccines [3]. The minimum sample size for a representative sample is calculated to be 1200, assuming a power of 80% at a confidence level of 95%. A population-based telephone survey will be conducted by randomly selecting the telephone numbers listed in the up-to-date telephone directories of HK residents. This will be conducted by external services provided by local companies. After the completion of the survey, 15–20 respondents will be selected using stratified sampling to achieve a representative sample with respect to age, gender, cultural/ethnic background, and economic background. These participants will be invited to attend an online in-depth qualitative interview to explore their reasons for vaccine hesitancy and thoughts/concerns in relation to COVID-19 vaccines. More participants will be recruited in this way until data saturation is reached. The findings from this mixed-methods study will provide valuable information for the development of the ‘AI-driven Vaccine Communicator’ and enable us to ascertain the extent of hesitancy.

Development of the ‘AI-driven Vaccine Communicator’. The development of the psychoeducation programme will be guided by the Vaccine Hesitancy Determinants Matrix Model (which incorporates vaccination-specific, individual/group, and contextual influences on attitudes towards vaccines) and will integrate MI skills. In addition, educational information regarding the individual, group, and contextual aspects of vaccine hesitancy will be provided on the Web-based intervention platform. The materials on our Web-based platform will be available in multiple formats, including text, videos, and animations, explaining (a) how the vaccine works against COVID-19 and (b) the process of research and development of COVID-19 vaccines and how their safety is ensured. Importantly, we will design a human–computer interaction system, the AI Digital Assistant with MI skills (available in Mandarin, Cantonese, and English), using artificial intelligence-based natural language processing technology. After the initial development of the ‘AI-driven Vaccine Communicator’, the programme content will be validated by a panel of experts in infection control, vaccine science, and MI-based psychosocial interventions. A small-scale pilot test with 8–10 participants will be performed to test the initial feasibility of the programme and gather valuable feedback for improving the content. We will continue to refine the system whenever new information or ideas emerge regarding any questions/concerns in relation to COVID-19 vaccines from our study participants.

Stage 2: Participants of stage 2 will be identified from the survey population of study 1. All Hong Kong-resident adults with COVID-19 vaccine hesitancy will be eligible. People with self-reported cognitive impairments or mental disorders that might interfere with their understanding of the intervention content, and those currently engaged in another clinical trial in relation to COVID-19 vaccines, will be excluded. We calculated our sample based on sample size determination methods for repeated measures. Assuming a Type I error rate of 0.05, primary hypothesis of time x intervention interaction, a change of 1.0 as minimum difference in the mean values of the variable, and estimated correlations among repeated measure of the variable as 0.6 for T0–T1, 0.5 for T0–T2, 0.4 for T0–T3, and a statistical power of 80%, as well as 20% attrition rate, the sample size is calculated to be 168 (i.e., 84 for each group) for two study groups. Block randomisation with a size of 6 will be performed with a computer-generated list of numbers known only to an independent administrator.

Stage 3: The ‘AI-driven Vaccine Communicator’ will be disseminated through collaborating clinicians, NGOs, community centres, and public/private healthcare providers (e.g., GPs). The five main components of the service evaluation assessment will include (1) reach—the proportion of approached participants expressing hesitancy towards COVID-19 vaccines and engaging with the programme throughout the study; (2) adoption—the proportion of participants adopting/completing the Web-based psychoeducation programme; (3) effectiveness—average outcome scores and consistent improvement in vaccine hesitancy outcomes; (4) implementation—the consistency of delivery of the programme, and effect of adaptations made to the implementation process and cost of the delivery; and (5) maintenance—the intention of the research team and participants to continue or adapt the intervention. All relevant community-based/clinical parties will be encouraged to apply our programme in their usual practice and work, our research team will trace all feedback from participants and all participants will be contacted again at 3-month intervals to assess their perceptions of and actions pertaining to COVID-19 vaccination. Stage three (the service evaluation) will consider the practical implementation of our intervention to a wider population, including its reach. Therefore, we will not limit the number of participants for stage 3 because one of the important outcomes is to ascertain how many people access and use it over the evaluation period. The estimate of minimum participant numbers for stage 3 is based on practical issues rather than statistical power to detect significant differences.

#### 10.2.4. Data Types, Processing, and Analysis

Stage 1. The participants’ age, gender, marital status, education and current occupation, and their intention to be vaccinated against COVID-19 (or receive booster doses), will be assessed using the WHO SAGE 10-item Vaccine Hesitancy Scale and the 4-Item Vaccine Confidence Index, for which valid and reliable versions are available in Chinese (both Mandarin and Cantonese) and English. After the first wave of surveys has been completed, 20–3 respondents will be selected using stratified sampling to achieve a representative sample with respect to age, gender, cultural/ethnic background and economic background, in that order. These participants will be invited to attend an online in-depth qualitative interview to explore their reasons for vaccine hesitancy and thoughts/concerns in relation to COVID-19 vaccines as identified in the initial cross-sectional survey. More participants will be approached for interviewing until data saturation is reached. All of the survey instruments will be scored according to the published guidelines and summarised using descriptive statistics, and univariate analysis will be performed to compare different vaccine acceptance/hesitancy-related outcomes among respondents with various socio-economic backgrounds. Qualitative content analysis guided by the Vaccine Hesitancy Determinants Matrix Model will be performed to comprehensively analyse the reasons for vaccine hesitancy among the HK population. The video-recorded interviews will be transcribed into text (either in Chinese or English). The data will be content-analysed and coded into categories. Any discrepancy in coding/category will be resolved by discussion, and the final themes/subthemes will be agreed upon by our team. The reflexivity and trustworthiness of the data will be guaranteed through in-depth interviewing techniques, and through rigorous data analysis involving more than three researchers.

Stage 2. Primary outcome: Vaccine hesitancy measured by adult Vaccine Hesitancy Index; Secondary outcome: Vaccine confidence assessed by Vaccine Confidence Index; A question with a ‘Yes/No’ answer will be asked: Have you scheduled a specific date for the next COVID-19 Vaccination? An open-ended question about people’s perception of and behaviour pertaining to COVID-19 vaccines will be asked. We will also assess participants’ perception of their degree of attitudinal change using three one-item measures of readiness for vaccine perception: How important is it for you to receive a COVID-19 vaccine?; How ready are you to receive a COVID-19 vaccine?; and How confident are you about receiving a COVID-19 vaccine? The participants will answer these questions at each follow-up on a discrete 11-point scale from 0 (not important/ready/confident at all) to 10 (highly important/ready/confident). Outcomes will be assessed at pre-, post-, 3- and 6-month follow up, and booster sessions will be provided every three months to further enhance their motivation for COVID-19 vaccine uptake throughout the study period. Data Analysis: All quantitative data from the pre-and post-tests will be numerically coded, summarised (with descriptive statistics), and analysed. The generalised estimating equation will be used to analyse the effects of the intervention. A tracking system will be implemented to examine the participants’ compliance in using the programme, e.g., the number of views and duration of time spent engaging with the programme. After the first follow-up assessment, selected participants (those with or without improvements in the study outcomes) will be invited for an in-depth qualitative interview to explore their thoughts on the strengths and limitations of the Web-based psychoeducation programme tested in the RCT and will be asked a set of questions assessing the usability of the programme. Any further amendments to the intervention and research procedures will be made based on the feedback from the participants’ interviews. The knowledge base of the programme and its technical foundation will be continuously consolidated and refined during the programme development phase and thereafter when feedback on the programme is available from the participants.

Stage 3. We will not have a control group but will use pre-post designs as our primary focus is the practical implementation of the intervention to a wider population. The statistical analysis methods for effectiveness outcomes of stage 3 will be ANCOVA for continuous variables and Chi-square tests for categorical variables. Our programme ‘AI-driven Vaccine Communicator’ will be disseminated to relevant stakeholders in healthcare settings and the general population to enable its use among a wider population. Considering the likelihood that new COVID-19 vaccines will be developed in response to viral mutations and that the vaccination programme will be expanded to other age/priority groups, our programme will be updated and adapted to sustain its long-term significance and usability.

## 11. Study 5

An educational intervention using a virtual reality game to improve compliance with COVID-19 disease-related hygiene practices and participation in SARS-CoV-2 screening, for the prevention of its transmission between primary schoolchildren in communities: a large-scale stratified cluster RCT.

Given the need to prevent COVID-19 transmission by this age group, an educational VRG should be implemented for primary schoolchildren, to improve their hygiene compliance and increase their awareness of community testing. This study will use a CBPR framework focused on equal participation between the researchers and the community stakeholders being studied (i.e., schools, teachers, schoolchildren and parents).

### 11.1. Aims

The proposed project will aim to investigate the ability of a virtual reality game (VRG)-based educational intervention to enhance primary schoolchildren’s compliance with hygiene practices and increase their participation in SARS-CoV-2 screening, to prevent SARS-CoV-2 transmission in the community. The objectives are as follows.

To assess the ability of a VRG-based educational intervention to enhance the knowledge and hygiene-practice compliance of primary schoolchildren in terms of the prevention of SARS-CoV-2 transmission, in comparison with a control group receiving usual community care, at 6 months after the intervention.To compare the bacterial loads on primary schoolchildren’s hands between the study group and the control group at 6 months after the intervention.To compare the detection rates of SARS-CoV-2 and other respiratory viruses in primary schoolchildren’s throat gargle samples between the study group and the control group at 6 months after the intervention.To compare the number of schoolchildren who participate in the community SARS-CoV-2 screening programme between the study group and the control group at 6 months after the intervention.

### 11.2. Plan of Investigation

This will be a single-blinded, cluster RCT with two-armed repeated measures.

#### 11.2.1. Subjects

Primary schoolchildren will be recruited from local primary schools. The eligibility criteria will be (1) primary school children aged 6–12, and (2) able to communicate in Chinese and to read Chinese. The exclusion criteria will be (1) reported mental or physical health disorders and (2) engagement in other health educational programmes related to hygiene practices 2 months or less before recruitment. The school management staff will help us to liaise with the schoolchildren and refer them to the research team. Informed consent will be obtained from school principals, parents and schoolchildren before the study begins. To minimise selection bias, the children will not be informed of whether they will join the intervention group or the control group. Potential subjects will be screened and recruited in these schools by the research team. In order to avoid data contamination in the control group, school children who have siblings in the intervention group will not be recruited. Details of the study will be provided to the schools, the schoolchildren and their parents. Sample size has been computed to obtain a statistical power of no less than 0.8 to detect the significance of the four objectives with continuous outcomes at a 0.05 level of significance. With 0.2 as the Cohen’s d effect size and under balanced allocations, we have set the intra-cluster correlation to 0.05, the correlation among level-2 units nested in a cluster to 0.01, the number of level-1 units to 30, and the sample size of level-2 stratified units to 2, we have 18 school clusters in each arm. Therefore, the sample size should be 2160.

#### 11.2.2. Methods

##### Phase 1: Pilot Study (8 Months)

The team will further refine the VRG educational programme based on the results and feedback from our previous pre–post intervention study. Then, two schools will be invited to participate in the pilot study. Training in student recruitment will be provided to the teachers. The recruited schoolchildren will participate in the VRG educational programme, which will be delivered by the researchers, for 6 months. Parents will also receive training in the assessment of their children’s hand-hygiene practices and collection of throat gargle samples at home. Feedback will be collected before (T1) and immediately after the intervention (T2) from the teachers, parents and schoolchildren on the feasibility of subject recruitment, the understandability of hygiene practice assessment and the attractiveness and learnability of the VRG design. The intervention protocol will be refined according to the feedback collected from community stakeholders (schools, schoolchildren and parents).

To study the cost-effectiveness of the VRG, the VRG will be implemented in one group of 60 students and conventional educational programme will be implemented in another group of 60 students. The total costs spent on both groups will be calculated, these include material production, human resources, facilities and equipment for programme implementation. The effectiveness of the VRG and conventional education programme will be measured as the number of students demonstrating compliance to hand hygiene practice. The ratio of cost per unit of effectiveness can be calculated by dividing the total cost with the number of students demonstrating hand hygiene compliance. The ratio will be compared between the VRG programme and conventional education programme.

##### Phase 2: Main Study (40 Months)

Intervention group: Educational intervention using a VRG over 3 weeks.

The schoolchildren in the intervention group will take part in an innovative VRG-based educational programme featuring: (1) Web-based health education on (i) SARS-CoV-2 transmission, (ii) proper hand and respiratory hygiene practices and (iii) the importance of community SARS-CoV-2 screening and throat-gargle collection procedures; (2) built-in instant feedback on their performance of hygiene practices; and (3) a chat function for efficient nurse–parent–schoolchild interactivity and communication. The schoolchildren will receive three sessions (one per week) of the VRG-based educational programme, which will be provided by the research team during extracurricular activity time. Each session will consist of two VRGs and will last 30–45 min. Six VRGs will be designed, representing six ascending levels. After completing each level, schoolchildren will be granted access to the next level. This design will engage the schoolchildren in playing the VRGs. Levels 1 and 2 will address the fundamentals of SARS-CoV-2 transmission and hand hygiene; levels 3 and 4 will consist of family-based scenarios exemplifying the chain of infection and throat gargle collection; and levels 5 and 6 will address how to combat germs in school-based scenarios and the importance of community SARS-CoV-2 screening. The schoolchildren will be required to complete electronic questionnaires pre-post and 3 months and 6 months after they participate in the educational programme. Parents will use the online checklist provided to assess schoolchildren’s performance on the return-demonstration of hand hygiene and throat gargle pre–post, and 3 months and 6 months after the educational programme. To ensure the accuracy of the recordings of the schoolchildren’s hygiene practices, the return-demonstration will be video-recorded and uploaded to the online platform by parents, for analysis by the research team.

Control group. The schoolchildren in this group will receive the usual health education or care available in the community. The VRG equipment will not be made available to the schoolchildren in this group within the intervention period.

#### 11.2.3. Study Design

The duration of each schoolchild’s involvement in the study will be 6 months, with assessments at baseline before the intervention (T1), immediately after the intervention (T2), at 3 months after the intervention (T3), and at 6 months after the intervention (T4). It is proposed that 12 schools will participate in the study every 12 months, and thus it will take 36 months to complete the intervention in 36 schools. Four months will be used for data cleaning, analysis and report writing. Data will be collected at four time-points, T1, T2, T3 and T4, to test the sustained intervention effect.

Randomisation: A large-scale stratified cluster RCT will be undertaken. Primary schoolchildren will be recruited via local conventional school clusters from 18 districts in Hong Kong, which have been chosen to provide a geographically representative sample. The cluster sizes will be equal across school clusters. We will select 36 primary schools and will randomly assign them to the intervention group or the control group in a ratio of 1:1. To adjust for age as a confounding factor, the schoolchildren from each school cluster will be separately drawn from lower grades (Primary 1 to 3) and higher grades (Primary 4–6). Therefore, we have designed a three-level hierarchical mixed model that is randomised at the third level. Here, a school is considered to be a level-3 cluster, the grades within a school are considered to be level-2 stratified units and the schoolchildren in our trials are considered as level-1 units [58].

#### 11.2.4. Data Types

The instruments to be used in this proposed study include a demographic data sheet, an electronic questionnaire (e-questionnaire), an online checklist, and equipment for counting bacteria on hands and for respiratory virus testing. The information collected by the demographic data sheet will include the schoolchildren’s age and medical history. The e-questionnaire will measure the schoolchildren’s hand-hygiene compliance and awareness of hand-hygiene practice, and their participation in SARS-CoV-2 screening. The online checklist will record the schoolchildren’s hand-hygiene practices and their performance in gargle collection. The bacterial counts will serve as a measure of the bacterial loads on the schoolchildren’s hands. The results of respiratory virus testing will indicate the rates of upper respiratory tract infection in the schoolchildren.

##### Hand-Hygiene Compliance (T1, T2, T3 and T4)

A hand-hygiene compliance checklist [59] will be used to measure the extent of hand-hygiene compliance in the intervention and control groups. Higher scores indicate better performance.

##### Awareness of Hand-Hygiene Practice (T1, T2, T3 and T4)

Schoolchildren’s knowledge of hand-hygiene practice will be measured. To profile their prior knowledge and their level of understanding after the programme, 10 ‘true or false’ questions on hand hygiene adapted from the American Cleaning Institute in 2011 will be administered to both the intervention and control groups. This instrument presents a good content validity index of 0.86 and a good two-week test–retest reliability of 0.75 [60].

Schoolchildren’s rate of participation in SARS-CoV-2 screening (T1, T2, T3 and T4)

The number and percentage of children participating in SARS-CoV-2 screening, and their reasons for doing or not doing so, will be collected.

Bacterial loads on hands (T1, T2, T3 and T4)

The research team will collect swabs of the schoolchildren’s hands to calculate the total bacterial counts of the intervention and control groups. Collections will be performed after school hours.

Reverse transcription–polymerase chain reaction testing for SARS-CoV-2 and other respiratory viruses (at T1, T2, T3 and T4, and at any time schoolchildren exhibit a fever with upper respiratory-tract infection symptoms)

Gargle samples will be collected from the children for RT-PCR tests for SARS-CoV-2, influenza A and B viruses, respiratory syncytial virus, rhinoviruses and adenoviruses. Testing for SARS-CoV-2 in gargle samples will be out-sourced to a company or organisation that provides a high-quality service.

#### 11.2.5. Data Analysis

Descriptive statistics will be used to present all of the demographic, programme attendance and outcome data. Chi-square or independent sample *t*-tests (for categorical and interval data, respectively) will be used to identify the heterogeneity of demographic and baseline data in the two groups and settings. The effectiveness of the VRG-based educational intervention on the primary outcomes (the level of knowledge and practice, the bacterial loads and the detection rates of respiratory viruses) will be examined by linear mixed-effect modelling to compare changes in the primary outcome values of the intervention and control groups over time (from baseline to the two post-tests) on an intention-to-treat basis, using repeated-measures analysis of covariance, multivariate analysis of covariance and/or mixed modelling techniques.

## 12. Across All Studies, Other Common Elements Include

Stop-and-go criteria [39] will be used in terms of subject recruitment (<20% vs. >50% meeting eligibility criteria, respectively); intervention acceptability (<30% vs. >60 or >70%, respectively, in different studies), and loss to follow up (<30% vs. >70 or >80%, respectively, in different studies). Furthermore, the RE-AIM framework [61] will be used to guide the evaluation of all studies. We will collect metrics on Reach (e.g., number of website visits), Effectiveness (changes in stay-at-home motivation and intention or changes in hygiene practice), Adoption (e.g., the amount of content created by the public with the campaign hashtag), Implementation (e.g., consistency in implementation of the educational VRG), and Maintenance (e.g., number website visits after the programme) as appropriate, to document formative and summative impact. All analyses will be performed using SPSS 26.0 software and a *p* value < 0.05 will be considered statistically significant.

## 13. Ethics Considerations

Ethics approvals for each study has already been obtained from the Institutional Review Board of the Hong Kong Polytechnic University. Online or face-to-face written informed consent (and accent from minors) will be obtained from all participants before data collection. Agreements from the recruitment sites to carry out the research and co-participate will be obtained too. Participation is entirely voluntary, and subjects can withdraw from the research at any time without any penalty. Participants’ identities and data will be protected by maintaining anonymity and confidentiality. Only authorized personnel will be able to access the data for analysis. Mental health service information will be integrated into the intervention if participants feel anxious or have other emotive reactions/needs. In one study where employers are involved as subject recruiters, more specific processes have been established to prevent coercion and exertion of power. Study team members and collaborators will strictly abide by all principles of research ethics, protection of participant anonymity and their data.

## 14. Implications

The proposed studies are expected to have direct and positive impacts on the HK community and to reduce the risk of COVID-19 transmission by improving preventative and early detection measures, increasing compliance with and improving the experience with social distancing measures, promoting the uptake of vaccines and increasing eHealth literacy related to COVID-19. The results of these studies will also inform the planning and preparation of responses to future pandemics. Decreased infection rates are expected due to improved preventative behaviours and increased vaccine uptake. The long-term sustainability of the approach will be achieved through the CBPR model processes.

## 15. Conclusions

COPAR for COVID utilises a community-based participatory research approach in the proposed five studies to address COVID-19 reductions in the community from a series of experiments. In this unique model, all studies will incorporate an intervention development phase in conjunction with key community stakeholders, a feasibility study and an execution stage. Theory-driven interventions will address each study’s focus (e.g. social distancing, promotion of vaccine uptake, eHealth education, preventive measures and early detection). Decreased infection risks are expected due to improved preventative behaviours and increased vaccine uptake. Improvements are expected to be seen in the outcomes of the general population, especially in vulnerable and high-risk groups. Long-term sustainability of the approach will be achieved through the CBPR model.

## Figures and Tables

**Figure 1 ijerph-19-13392-f001:**
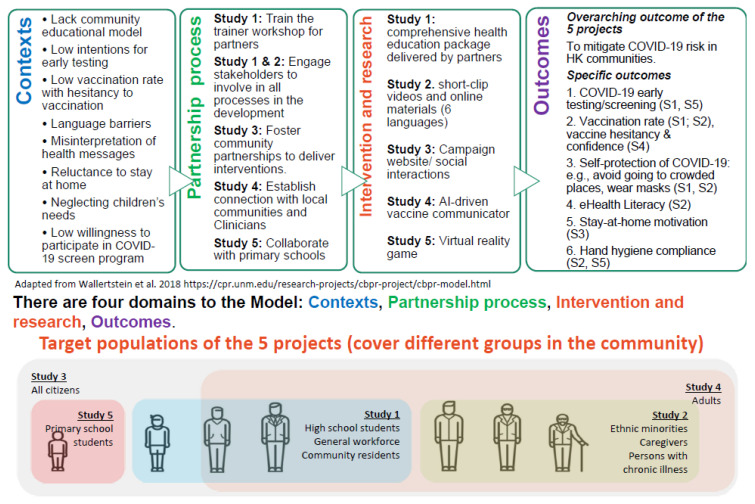
An adaptive CBPR model regarding to the COVID-19 epidemic context in Hong Kong. Adapted from: https://cpr.unm.edu/research-projects/cbpr-project/cbpr-model.html.

**Figure 2 ijerph-19-13392-f002:**
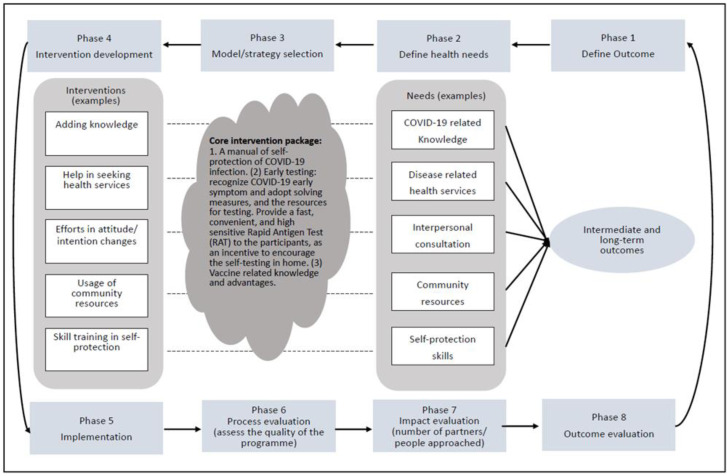
The adaptive PROCEDE-PROCEED model for designing and implementing the community-based education programmes in the “AID” project (study 1). Adapted from: [34]. Green L, Kreuter M. Health program planning: An educational and ecological approach. 4th edition/New York, NY, USA: McGrawhill 2005.

**Figure 3 ijerph-19-13392-f003:**
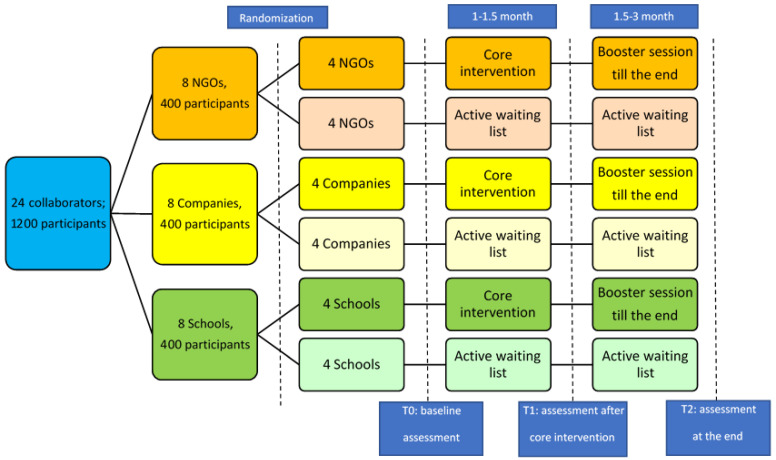
Flow chat of stage 3, the community-based interventions (Cluster RCT design) [Study 1].

**Figure 4 ijerph-19-13392-f004:**
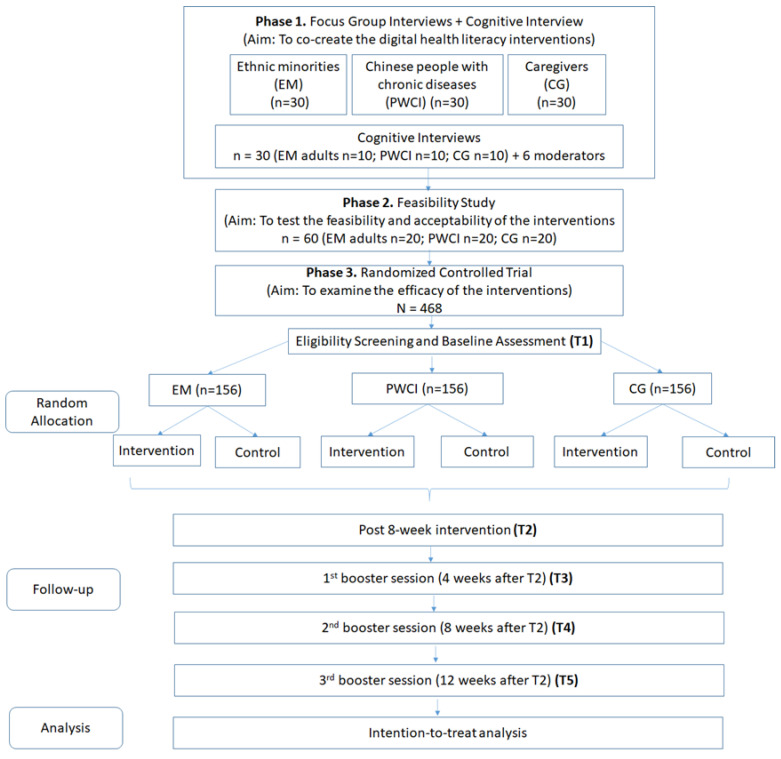
Flow chart of data collection of Study 2.

## Data Availability

The datasets that will be used and/or analysed during the current study will be available from the corresponding author on reasonable request.

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
