# Peer review of "A Community-Based Participatory Research Approach to Developing and Testing Social and Behavioural Interventions to Reduce the Spread of SARS-CoV-2: A Protocol for the ‘COPAR for COVID’ Programme of Research with Five Interconnected Studies in the Hong Kong Context"

_ijerph, 2022, doi:10.3390/ijerph192013392_

Round 1

Reviewer 1 Report

This a comprehensive and well-designed study protocol to reduce the spread of SARS-CoV-2 in the Hong Kong area by the increase early testing for COVID-19, promotion of COVID-19 vaccination, increase of health literacy and increase of preventive strategies such as hygiene and restriction of social contacts.

In the abstract, the authors state that “A variety of self-reported and objective- 30 based measures will be used to assess various outcomes, based on the focus of each study, in both 31 the short- and long-term”, can the authors name a few specific examples?

The study design is very complex, a graphical abstract to give an overview about participants, studies, aims, steps, and outcomes of the whole project would be helpful

Please use terms in a consistent way (sector, group, cluster …) and define them when first mentioned

The authors state that “An agent-based simulation model that considers human mobility in HK will be developed. The model will be trained using data from the 24 community-based intervention programmes, to establish the most optimal community-based health education model(s) for COVID-19 prevention.” Please explain this in more detail, it does not become clear how the optimal model is chosen.

Since we have end of August 2022 by now, how are the current vaccination rates, is it still possible to achieve relevant improvement within the expected timeline? What is the current stage of the study? Are the studies performed in parallel or sequentially?

Always clearly state, for which outcome which phase is powered

The manuscript still has typos, it should be proofread by a native speaker

Author Response

Please see attached file for both reviewers' comments

Reviewer 2 Report

The researchers have stated highly important goals of aiming to substantially improve the possibility of managing the present as well as future pandemics. This is a worthy cause that should be encouraged. The literature review is comprehensive, written in a logical flow, and productive for the readers. Nonetheless, the main focus of the article is to describe five studies that the researchers plan to implement. As these have not as yet been launched, and no results of their effectiveness exist to date, there is little worthiness to the readers at this point. The description of each study is exhaustive but as it's unclear at this stage whether any or all of the studies will prove to impact the compliance of the population, at this stage, I do not see the value of publication.

I suggest the authors expand the literature review and submit the article as a review article. In this format, they may brief delineate the proposed studies, but as a minor component of the article.

Author Response

Please see attached file (same as previous file, as comments from both reviewers were presented together in a single file)

Round 2

Reviewer 1 Report

The authors adequately responsed to all points raised by the reviewers, there are just a few typos in the new text which need to be corrected. 

Author Response

A few typos and grammar mistakes were corrected, after a careful proof-reading of the whole paper.

Reviewer 2 Report

I thank the authors for explaining the rationale of the publication at this early stage of the study. I'm not sure if publication of the study design will prompt other scientists to initiate such studies at this period of time, prior to any results that can present effectiveness. Nonetheless, the authors have convinced me that it is 'worth a try' and we will all learn from this, so that we'll have better insight in the future whether sharing of study designs elicit timely research initiatives.

My only comment is that this rationale should be clearly stated, both in the abstract and in the manuscript itself 

Author Response

Thank you for your comment, which we have added both in the abstract and in the main body of the paper (lines 294-298) as requested.